# Detection and Factors That Induce *Stenocarpella* spp. Survival in Maize Stubble and Soil Suppressiveness under Tropical Conditions

**Felipe Augusto Moretti Ferreira Pinto** [1,*](https://orcid.org/) **, Victor Biazzotto Correia Porto** [2]**, Rafaela Araújo Guimarães** [3]**,
Carolina da Silva Siqueira** [3] **, Mirian Rabelo de Faria** [4]**, José da Cruz Machado** [3]**, Henrique Novaes Medeiros** [3]**,
Dagma Dionísia da Silva** [5]**, Helon Santos Neto** [3]**, Edson Ampelio Pozza** [3]
**and Flávio Henrique Vasconcelos de Medeiros** [3,*]

1   Empresa de Pesquisa e Extensão Rural-EPAGRI, Estação Experimental de São Joaquim,
    São Joaquim 88600-000, SC, Brazil
2   Fundação de Apoio à Pequisa e Desenvolvimento do Oeste Baiano—Fundação BA,
    Luiz Eduado Magalhães 47850-000, BA, Brazil; victorbcporto@gmail.com
3   Department of Plant Pathology, Universidade Federal de Lavras, P.O. Box 3037, Lavras 37200-900, MG, Brazil;
    rafaela.guimaraes3@ufla.br (R.A.G.); kerolpet@gmail.com (C.d.S.S.); machado@ufla.br (J.d.C.M.);
    medeirosxd@gmail.com (H.N.M.); helonareado@hotmail.com (H.S.N.); eapozza@ufla.br (E.A.P.)
4   FCA/Campus Botucatu, Universidade Estadual Paulista Júlio Mesquita Filho, Botucatu 186108-034, SP, Brazil;
    mirianrabelofaria@yahoo.com.br
5   Embrapa Milho e Sorgo, P.O. Box 285, Sete Lagoas 35701-970, MG, Brazil; dagma.silva@embrap.br
*   Correspondence: felipepinto@epagri.sc.gov.br (F.A.M.F.P.); flaviomedeiros@ufla.br (F.H.V.d.M.)

**Abstract:** *Stenocarpella* spp. causes stalk and ear rot in maize and overwinters in stubble during the off-season. Understanding the factors that guide saprophytic colonization is a crucial strategy for management. In this study, we analyzed the abiotic factors and crop management practices in relation to the inoculum of *Stenocarpella* spp. in stubble by qPCR. Soil samples were used for suppressiveness tests against *Fusarium verticillioides*, *Fusarium graminearum*, and *Stenocarpella maydis*. In the 29 fields, different levels of *Stenocarpella* spp. were detected. Only three fields were considered suppressive for the three pathogens. Heat maps showed that soil suppressiveness was inversely related to the pathogen concentration, and the suppressiveness of one pathogen was correlated with the suppressiveness of other pathogens. Under no-tillage systems in which rotation with soybeans was adopted, *Stenocarpella* spp. were detected at lower concentrations than in areas that adopted no-tillage systems with maize monocultures. While in tillage systems, the maize–maize monocropping increases the inoculum level of *Stenocarpella* spp. Crop rotation is a factor related to the observed reduction in the pathogen concentration and increases in the broad-spectrum antagonistic microbial communities. These communities guide the suppressiveness of soil-borne diseases in maize fields cultivated under tropical conditions.

**Keywords:** *Zea mays*; stalk diseases; crop rotation; stubble; suppressive soils

## 1. Introduction

Maize (*Zea mays* L.) stalk and ear rot diseases result in significant post-harvest economic losses and reduce the quality and quantity of grains. Several fungi can cause stalk and ear rot, the most important being *Stenocarpella* spp. and *Fusarium* spp. [1]. These pathogens colonize the aerial parts of maize plants due to conidial dispersal by wind, water splashing, and contaminated seeds. Indeed, seeds disperse the inoculum of the pathogen over long distances, but there is no report of *Stenocarpella*-caused disease outbreaks in the first season in which maize is cultivated. In order for an outbreak to occur, the inoculum levels in the maize stubble need to reach sufficient levels [2].

*Stenocarpella macrospora* and *S. maydis* are the predominant species within the genus and can be collectively detected using a genus-specific primer [3]. A molecular biology approach based on qPCR (quantitative PCR) is an easy, fast, and sensitive test for specific targets than conventional methods. When more than one pathogen causes infection in a plant and requires accurate detection, qPCR is a suitable method for detection and quantification [4].

*S. macrospora* and *S. maydis* are seed-transmitted, cause stalk rot and ear rot, and *S. macroscopora* is associated with macrospora leaf spot. These diseases may not only compromise photosynthesis by the reduction in the green leaf area but also serve as a reservoir for the pathogen inoculum build up and infect the kernels causing ear rot or at least transmit the pathogen. These species can survive on seed and/or colonize the maize stubble. In this environment, the spores can increase or remain dormant until the following raining and planting season [5]. Such stubble also serves as a reservoir for other maize pathogens that compete for the same nutrients (stalk and kernels), and the most important are *F. verticillioides* and *F. graminearum* [6].

Conversely, maize stubble does not last long in tropical agriculture systems due to the speed with which such organic matter is decomposed. Therefore, there until now is no evidence of the role of the different maize stubble (in relation to size, crop system, and climatic conditions) in the survival of *Stenocarpella* app. on maize. Furthermore, *Stenocarpella* spp. is host-specific; therefore, the presence of a maize monoculture likely plays a detrimental role in the overwintering of the pathogen [7], especially in tropical agriculture systems, where the weather is favorable for volunteer maize plants to not only harbor but also build up inoculum of *Stenocarpella* spp. [8].

Within the integrated management of diseases, soil biological properties play an important role. It can influence the selection of antagonistic microbial communities associated with the pathogen reduction and elimination, resulting in the reduction in the disease. These soil conditions, the use of fungicides (chemical and biological), and the resistance of cultivars do not necessarily result in satisfactory disease control [9].

In Brazil, no-tillage cropping is the default practice, but it relies on the sequence of soybean and maize planted in a succession, not in rotation [10]. This system reduces the soil preparation time and maintains soil humidity, thus reducing erosion [11]. However, crop stubble on the soil surface supports the survival of many plant-stubble-borne pathogens, which are saprophytes and can cause disease outbreaks [12].

There is considerable knowledge on the epidemiology of other species that cause ear rot on maize plants, such as *F. graminearum* and *F. verticillioides* [13]. In contrast, knowledge of the occurrence and epidemiology of the *Stenocarpella* complex is scarce, particularly with regard to tropical agriculture. This knowledge is essential for the development of preventive measures to reduce the risk of fungal infections during the growing season. The objectives of this study were to identify the source of inoculum for *Stenocarpella*, the contribution of maize growing areas and crop rotation on the *Stenocarpella* spp., as well as to understand the suppressiveness of soil towards *S. maydis*, *F. verticillioides*, and *F. graminearum*.

## 2. Materials and Methods

### 2.1. Identification of the Inoculum Source for Stenocarpella spp. within Maize Stubble

To determine the major sources of inoculum, samples were taken in a maize field under a no-tillage management system in Lavras (Minas Gerais, Brazil) during the season of 2015/2016. The evaluated samples consisted of stalks, grains, cobs, decaying maize leaves, and dead weeds (species of weed were not identified). Each type of the different plant-derived material (e.g., stalks together) was used separately as a sample for the study of *Stenocarpella* spp. inoculum sources in the field sampling. For each type of sample, the DNA was extracted, and relative quantification of *Stenocarpella* spp. was conducted.

## 2.2. Classification of Decomposition on Maize Stubble

Once the maize plant part within the stubble that harbored the highest inoculum source was determined, a second evaluation was performed on the inoculum level according to the decomposition rate of the corresponding plant part. Maize stubbles were classified into three levels of decomposition: low, medium, and high decomposition of stubble. The low decomposition classified the stubble regarding fragmental size as entire. The medium decomposition was partially disintegrated, and the high decomposition was disintegrated.

## 2.3. Sampling in Maize Fields

After determining the maize plant part within the stubble that harbored the highest concentration of *Stenocarpella* spp. inocula, 29 samples of this plant part were collected in 15 different locations (Figure 1). The sampling system took into account the rotation system (whether the grower adopted crop rotation) and soil tillage (no-tillage or conventional tillage). The number of replicates and the sampling strategy were similar to those described above.

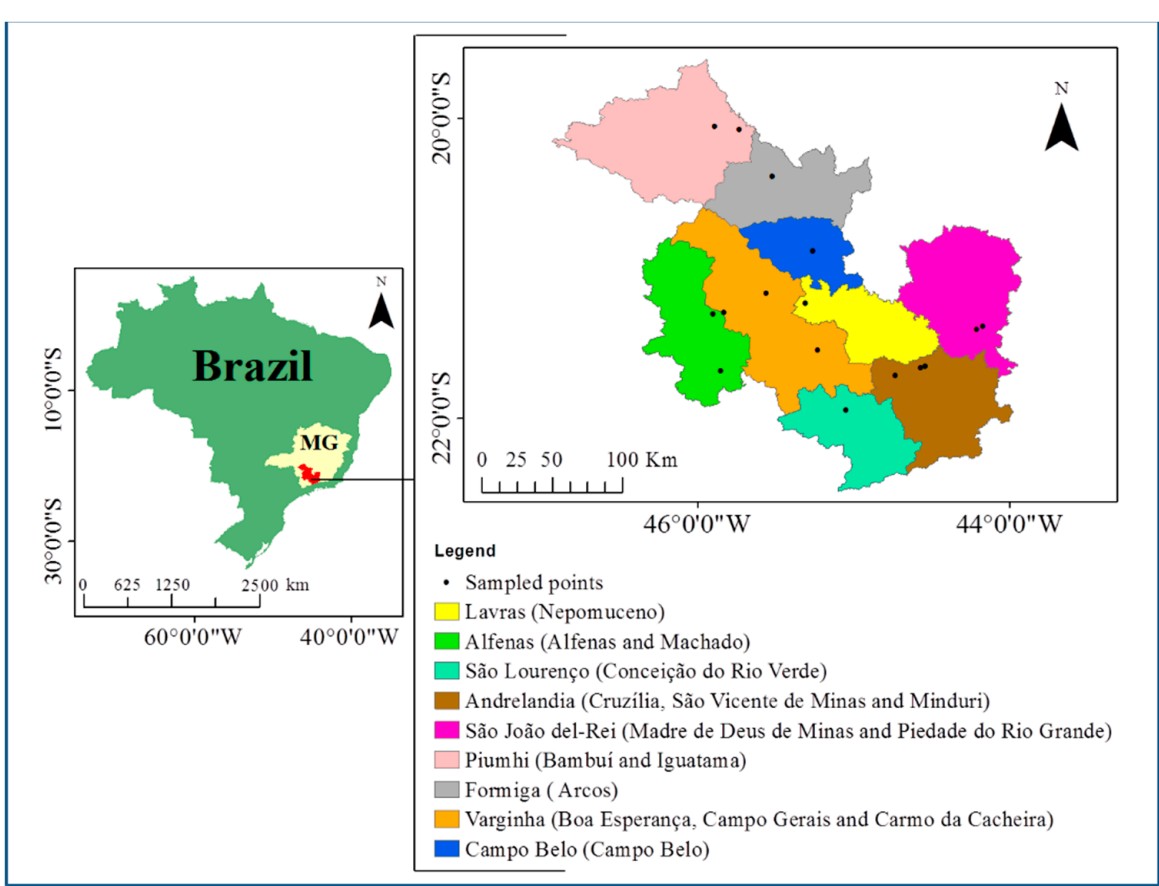

**Figure 1.** Location map of sampled areas in Minas Gerais state (MG), Brazil.

## 2.4. Sample Processing, DNA Extraction, and Quantitative Real-Time PCR (qPCR)

The samples were ground with a 1-mm mesh sieve, freeze-dried, and stored at −18 °C until DNA extraction was conducted. Approximately 40 mg of lyophilized sample were used for DNA extraction, employing a Wizard® Genomic DNA Purification Kit (Promega, Madison, WI, USA), following the protocol recommended by the manufacturer [14].

All DNA samples from the same area were pooled to determine the presence of *Stenocarpella* spp. in a given field. The qPCR analysis was conducted using an SYBR Green PCR Master Mix on a Rotor-Gene 6500 (Corbett Research, Mortlake, Australia). For each reaction, a 2.0-μL sample was mixed with 23 μL of reaction mix containing 12.5 μL SYBR Green PCR Kit (Qiagen, Hilden, Germany) and 0.75 μM of each forward

and reverse primer. The primers used were P1 (GTTGGGGGTTTAACGGCACG) and P2 (GTTGCCTCGGCACAGGCCGG), sequenced and designed from the ITS1 and ITS2 regions of rDNA as described by [3], which are specifically for the detection of the levels of DNA copies from *Stenocarpella* spp. in biological samples, according to [5]. A 5-fold serial dilution (20 ng to 0.002 ng of DNA) of *S. maydis* (isolate LAPS 698) was included in each run as a positive control and to calculate the number of DNA copies generated in the cycles.

The qPCR conditions were as follows: 95 °C for 3 min; 40 cycles of 94 °C for 30 s, 60 °C for 1 min, and 72 °C for 1 min; and, finally, 72 °C for 10 min to melt the double-stranded DNA. The specificity of the amplicons was confirmed with melting-curve analysis of the qPCR products at the different DNA concentrations. The threshold curve was calculated with the Ct value, which was determined as the number of cycles in which the fluorescence generated within a reaction crossed the threshold. The comparative Ct method was also used. Samples showing the lowest expression of each gene were used as calibration samples, and relative expression was measured using the relative standard curve method. The values obtained corresponding to the sample DNA levels were compared to the control DNA level. To calculate the gene expression levels, the following were considered: the Ct values (exponential increase in PCR product) of the target gene and endogenous control, $\Delta Ct = Ct$ (sample) $-$ Ct (endogenous control), and $\Delta\Delta Ct = \Delta Ct$ (sample) $- \Delta Ct$ (calibrator). The gene expression levels were then calculated using the formula RQ = 2 $- \Delta\Delta Ct$, where RQ means relative quantifications [14].

### 2.5. Suppressiveness against Soil-Borne Diseases

To identify the suppressiveness of soils from the different sample sites, we collected soils (0–5 cm depth) from the same sites where maize stalks of bulk soil were sampled (Supplementary Table S1). The suppressiveness was evaluated for the three pathogens that frequently overwinter on maize stubble and cause similar stalk and ear rot damage to maize: *F. verticillioides*, *F. graminearum*, and *S. maydis*. The methodology proposed by [15] was used with modifications. The soil samples were individually placed into Petri plates (9 mm), watered to 60% field capacity, and inoculated with 5 mL of $10^5$ conidia $mL^{-1}$ of each pathogen (*F. verticillioides*, *F. graminearum*, and *S. maydis*). Finally, the sampled soils were classified as suppressive (considerable bait colonization reduction), intermediate (some effect in bait colonization reduction), and conducive (considerable bait colonization increase), as proposed by [16].

### 2.6. Data Processing and Statistical Analysis

The experiments were carried out in a randomized block design (RBD). For statistical analyses, the Shapiro–Wilk test was performed for normalization. For data regarding the different types of stalks used in qPCR, the Ct values and the classification of decomposition on maize stubble were submitted to one-way ANOVA and Tukey's multiple comparisons of means at a 95% family-wise confidence level. The incidence data gathered in each replicate from the soil suppressiveness assay were submitted to variance analysis (ANOVA). For the evaluation of significant effects, the means were compared according to Scott–Knott's test ($p \leq 0.05$). Both analyses were conducted using the software R.

The relative quantifications (Ct values) and locations of the collected samples were submitted to an agglomerative hierarchical cluster analysis based on Ward's grouping method. A correlation matrix [17] was constructed to group the 16 different municipalities. The clustering was based on a dissimilarity matrix of Euclidean distances between individual municipalities. The same methods were also used for the sampled sites, and clusters were used to define categories for the creation of a map of the quantification of *Stenocarpella* spp. inocula. Thus, heat maps were created for the classes of the relative quantifications of *Stenocarpella* spp. inoculum through Pearson's correlation with the annual average temperature, annual average precipitation, altitude, and average maize yield. The heat maps were created using the inverse distance weighting interpolation method. The software used to build the maps was ArcGIS 10.3.

## 3. Results

Among the different maize parts found in the stubble, stalks were the only substrate encountered in all sampled areas that harbored *Stenocarpella* spp. However, *Stenocarpella* spp. was observed in higher numbers on grains (Ct = 26.86) and cobs (Ct = 25.08) when they were found in the main crop stubble. The decaying maize leaves (Ct = 32.21) and dead weeds (Ct not detected) did not harbor inoculum at high enough levels to be detected (Figure 2). Since stalks were the only maize found at all sampled sites, we decided to consider this plant part with which to make comparisons and inferences.

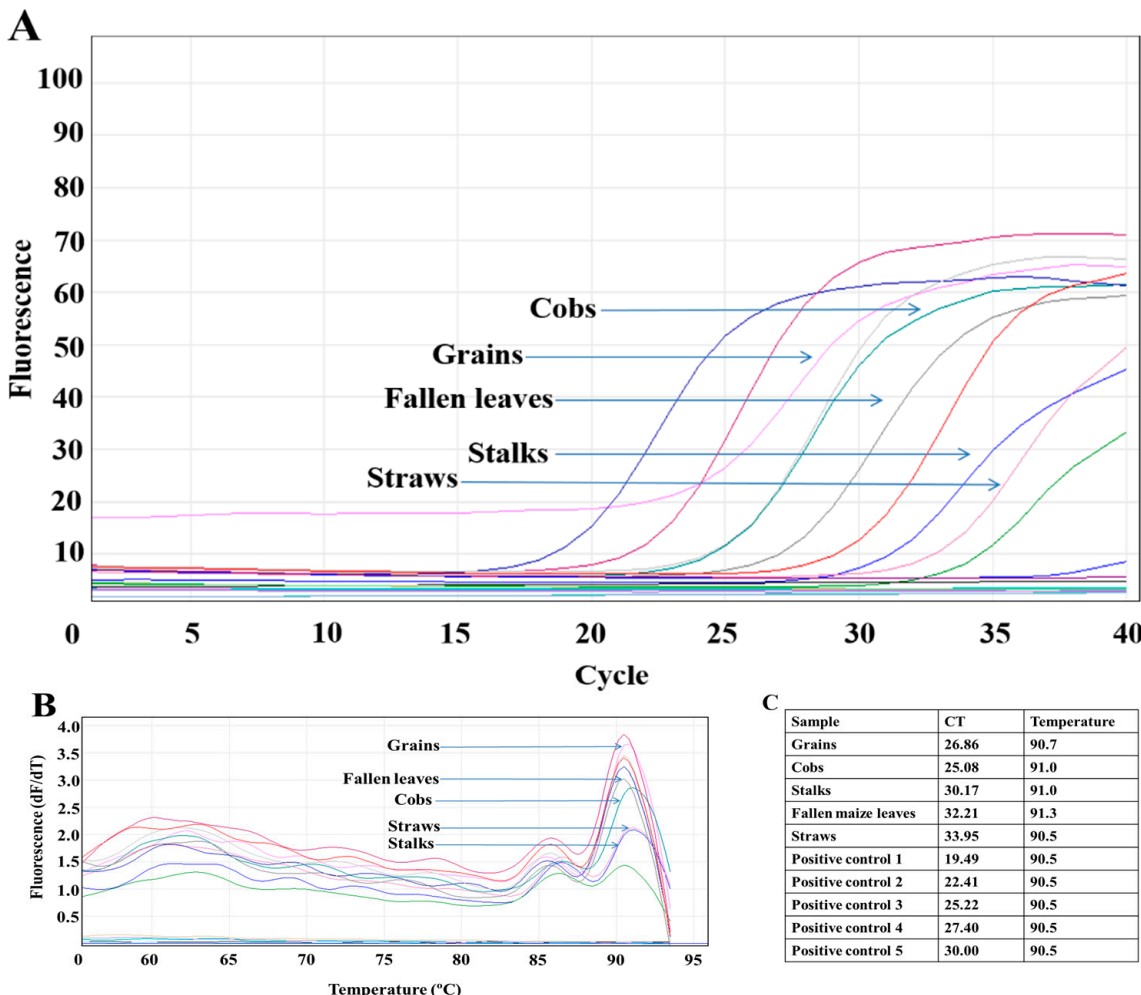

**Figure 2.** Melting curve analysis for *Stenocarpella* spp. using real-time polymerase reaction (PCR) assay. (**A**) The amplification curve of different substrates and controls obtained with primers P1/P2 [3]. (**B**) Melting curve analysis of the same samples shows the presence of the specific PCR product. (**C**) Comparison chart with cycle threshold (Ct) values and melting temperatures. Ct means the cycle threshold values. P1 = Primer used (GTTGGGGGTTTAACGGCACG) as reference for *Stenocarpella* spp. P2 = Primer used (GTTGCCTCGGCACAGGCCGG) as reference for *Stenocarpella* spp.

RQ was higher in fields (fields 15 and 24) that were cultivated with the adoption of conventional tillage practices. On the other hand, in sites managed under no-tillage systems, *Stenocarpella* spp. was always detected, almost always in smaller concentrations (20 of the total 27 fields that adopted no-tillage). However, in 7 fields (1, 16, 18, 20, 21, 25, and 29), RQ was higher. This value represents 25% of no-tillage fields. In the monocropping fields (fields 10, 11, 14, 17, 19, 21, 23, and 24), RQ levels were different, and only 2 fields (21 and 24) increased RQ levels (Table 1).

**Table 1.** Relative quantification (RQ) of *Stenocarpella* spp. (DNA copies) determined by qPCR in stubble and colonization of baits (%) by soil-borne pathogens (*Stenocarpella maydis*, *Fusarium graminearum*, and *Fusarium verticillioides*) in collected soils.

| Field [1] | Crop in Sequence [2] | Tillage [3] | Maize Stubble | Baits Colonization in Soils (%) | | |
|---|---|---|---|---|---|---|
| | | | RQ of *Stenocarpella* spp. | *Stenocarpella maydis* | *Fusarium graminearum* | *Fusarium verticillioides* |
| 1 | M-S | No | 1 | 95 a * | 85 b | 33 c |
| 2 | M-S | No | $8.49 \times 10^{-11}$ | 100 a | 100 a | 100 a |
| 3 | M-B-S | No | $6.06 \times 10^{-11}$ | 10 c | 14 c | 10 c |
| 4 | M-S | No | $7.10 \times 10^{-10}$ | 5 c | 24 c | 5 c |
| 5 | M-B | No | $1.61 \times 10^{-10}$ | 71 b | 62 b | 38 c |
| 6 | M-S | No | $1.31 \times 10^{-10}$ | 95 a | 95 a | 100 a |
| 7 | S-M-S | No | $3.93 \times 10^{-9}$ | 71 b | 100 a | 90 b |
| 8 | S-M-S | No | $8.59 \times 10^{-9}$ | 33 c | 67 b | 29 c |
| 9 | M-W-S | No | $4.87 \times 10^{-11}$ | 71 b | 62 b | 71 b |
| 10 | M-M | No | $6.34 \times 10^{-11}$ | 76 b | 95 a | 86 b |
| 11 | M-M | No | $1.58 \times 10^{-10}$ | 100 a | 100 a | 86 b |
| 12 | M-W-M | No | $3.90 \times 10^{-10}$ | 29 c | 29 c | 43 c |
| 13 | S-W-M | No | $9.80 \times 10^{-7}$ | 95 a | 95 a | 100 a |
| 14 | M-M | No | $7.89 \times 10^{-11}$ | 86 b | 71 b | 76 b |
| 15 | M-B | Yes | 1 | 38 c | 76 b | 33 c |
| 16 | M-S | No | 1 | 100 a | 100 a | 100 a |
| 17 | M-M | No | $1.70 \times 10^{-11}$ | 33 c | 100 a | 100 a |
| 18 | M-S | No | 1 | 95 a | 100 a | 95 a |
| 19 | M-M | No | $2.15 \times 10^{-12}$ | 100 a | 100 a | 100 a |
| 20 | M-S | No | 1 | 76 b | 100 a | 81 b |
| 21 | M-M | No | 1 | 86 b | 90 a | 95 a |
| 22 | M-S | No | $5.86 \times 10^{-12}$ | 100 a | 95 a | 100 a |
| 23 | M-M | No | $1.23 \times 10^{-11}$ | 100 a | 100 a | 95 a |
| 24 | M-M | Yes | 1 | 95 a | 100 a | 100 a |
| 25 | M-S | No | 1 | 100 a | 100 a | 100 a |
| 26 | M-M | No | $7.71 \times 10^{-12}$ | 100 a | 100 a | 100 a |
| 27 | M-S | No | $8.52 \times 10^{-11}$ | 81 b | 71 b | 71 a |
| 28 | M-S | No | $4.18 \times 10^{-8}$ | 95 a | 100 a | 100 a |
| 29 | M-S | No | 1 | 81 b | 86 b | 86 b |

[1] Field locations are available in Supplementary Table S1. [2] Crop in sequence: M-S (Maize-Soybean), M-B-S (Maize-Bean-Soybean), M-B (Maize-Bean), S-M-S (Soybean-Maize-Soybean), M-W-S (Maize-Wheat-Soybean), M-M (Maize-Maize), S-W-M (Soybean-Wheat-Maize), and M-B (Maize-Bean). [3] Tillage system: No (No-tillage) and Yes (Tillage). * Lowercase letters in columns indicate no statistical differences by Scott–Knott cluster analysis ($p \leq 0.05$).

According to one-way ANOVA and Tukey's multiple comparisons of means, stubble particle size contributed to pathogen survival, as measured by the DNA quantification of *Stenocarpella* spp. The relative quantification of stalks classified as entire was different from those classified as disintegrated and was equal to those classified as partially disintegrated. Areas with disintegrated stalks were different from areas with partially disintegrated stalks.

The rate of the relative quantification (RQ) of *Stenocarpella* spp. did not necessarily group according to the site location, i.e., neighboring municipalities. For example, Piedade do Rio Grande and Madre de Deus de Minas were grouped in different clades, and Campo Belo and Boa Esperança, which are neighboring sites, were not grouped together; therefore, it was not the location of the site but rather other factors that likely governed such inoculum levels (Figure 3).

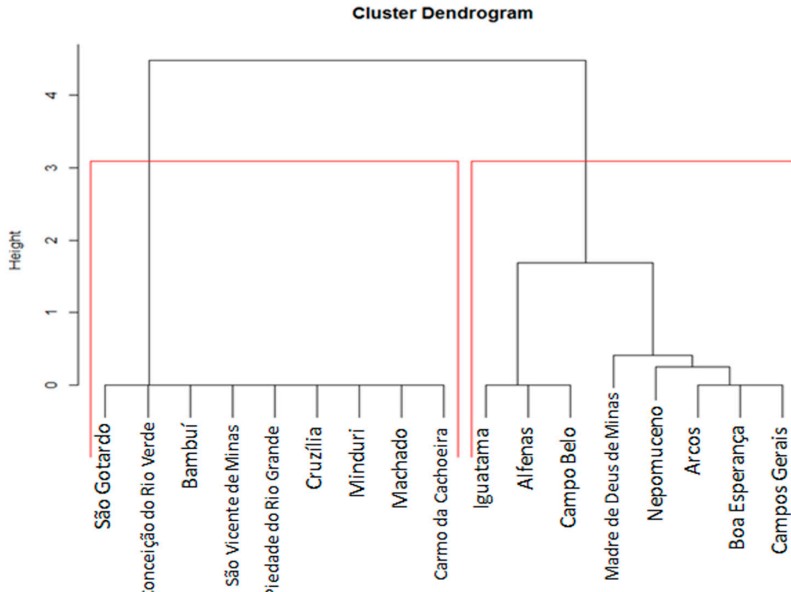

**Figure 3.** Cluster dendrogram of the DNA qPCR-obtained relative quantifications of *Stenocarpella maydis* by sampled location. The dendrogram was generated by data submitted to an agglomerative hierarchical cluster analysis based on Ward's grouping method and correlation matrix [17].

Based on the obtained results (Table 1), the sampled sites were classified into three groups for each pathogen, following these characteristics: suppressive (considerable bait colonization reduction), intermediate (some effect in bait colonization reduction), and conducive (considerable bait colonization increase). From the 29 sampled fields, only 3 fields (3, 4, and 12) were classified as suppressive (all were classified as 'c' in Scott–Knott cluster analysis) to all tested pathogens (Table 1). In addition, fields 3, 4, 8, 12, 15, and 17 were suppressive against *S. maydis* and fields 3, 4, and 12 were suppressive against *F. graminearum*. For *F. verticillioides*, fields 3, 4, 8, 12, 15, and 17 were considered suppressive. Whereas only 3 fields (5, 9, and 14) were classified as intermediate (all were classified as 'b' in Scott–Knott cluster analysis) to all tested pathogens. Thus, fields 5, 7, 9, 10, 14, 20, and 21 were intermediate against *S. maydis*, and fields 1, 5, 8, 9, 14, 15, 27, and 29 were intermediate against *F. graminearum*. For *F. verticillioides*, the fields 5, 7, 9, 10, 14, 20, 21, 27, and 29 were considered intermediate. While 13 fields (fields 2, 6, 11, 13, 16, 18, 19, 22, 23, 24, 25, 26, and 28) were classified as conducive (all were classified as 'a' in Scott–Knott cluster analysis) to the 3 pathogens tested. The fields 1, 2, 6, 11, 13, 16, 18, 19, 22, 23, 24, 25, 26, and 27 were conducive to *S. maydis*, fields 2, 6, 7, 10, 11, 13, 16, 17, 18, 19, 20, 21, 22, 23, 24, 25, 26, and 28 were conducive to *F. graminearum*, and fields 1, 2, 6, 11, 13, 16, 18, 19, 22, 23, 24, 25, 26, and 28 were conducive to *F. verticillioides* (Table 1).

Among the sampled soils, 45% were conducive to all 3 tested pathogens. Whereas 10% of these soils were suppressive, and another 10% were intermediate in reducing the inoculum of all evaluated pathogens. By considering the soil suppressiveness levels of the different sampled sites for *F. verticillioides*, *F. graminearum*, and *S. maydis*, a significant correlation was found (Table 2).

Finally, heat maps were generated based on the data of the *Stenocarpella* inoculum quantifications: numbers of maize stalk baits colonized by each of the pathogens (*S. maydis*, *F. verticillioides*, and *F. graminearum*), altitudes, rainfall levels, maize grain yields, and temperatures (Figure 4). The *Stenocarpella* inoculum map was designed based on the relative quantification of the pathogen in the stalks on soil and ranged from 0 to 3 (the closer to 0, the more inoculum and the closer to 3, the less inoculum of *Stenocarpella* ssp., as also suggested in Table 1). The lowest figures represent the highest concentrations of the pathogen. A pattern

of lower inoculum levels (green color) was observed on the edges of the evaluated area, and higher inoculum levels were observed in the middle of the study area.

**Table 2.** Pearson's correlation regarding of colonization baits in soils (%) and soil-borne diseases: *Fusarium verticillioides*, *Fusarium graminearum*, and *Stenocarpella maydis*.

| | Pearson's Correlation | | |
|---|---|---|---|
| **Colonization Baits (%)** | *Fusarium graminearum* | *Fusarium verticillioides* | *Stenocarpella maydis* |
| *Fusarium graminearum* | 1 * | 0.86 | 0.80 |
| *Fusarium verticillioides* | 0.86 | 1 | 0.78 |
| *Stenocarpella maydis* | 0.80 | 0.78 | 1 |

* The level of significance of evaluation is $p \leq 0.05$.

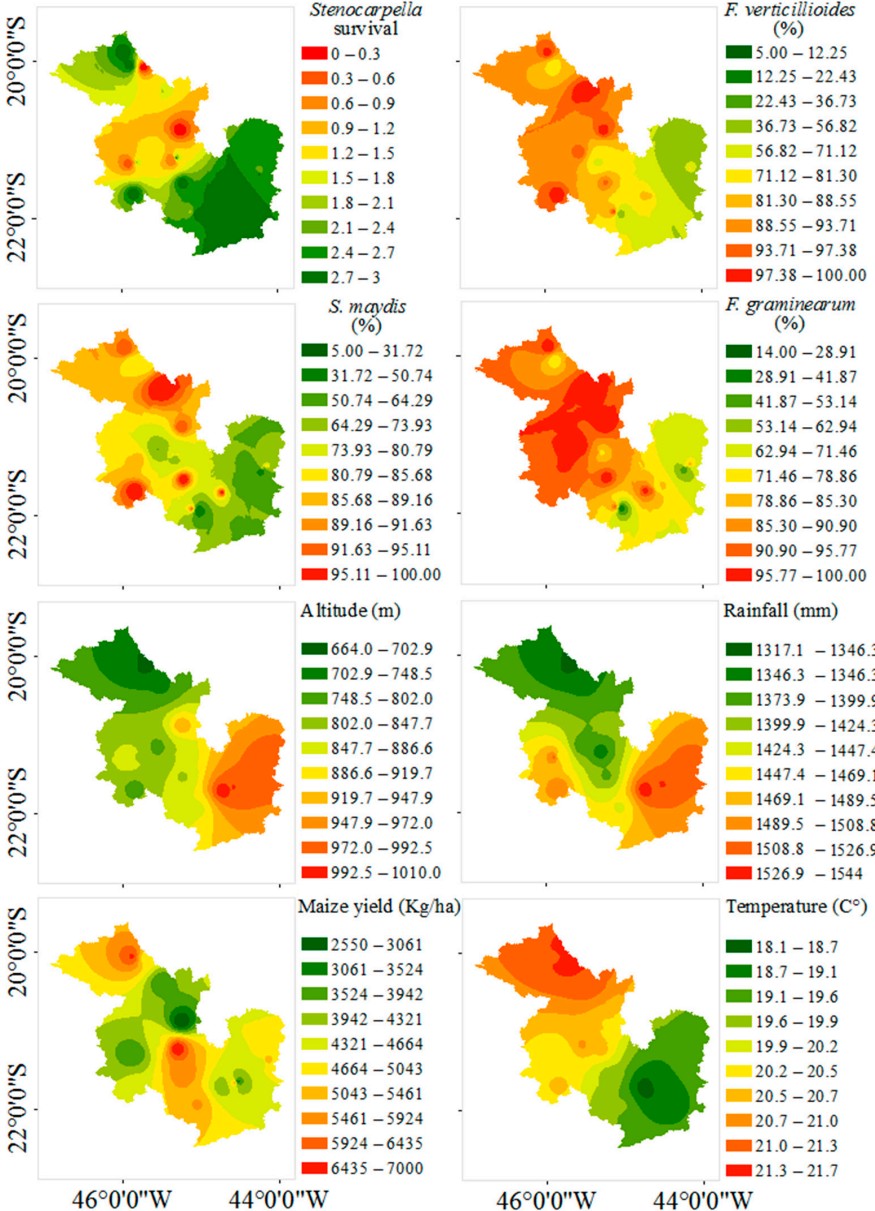

**Figure 4.** Heat maps of the classes of *Stenocarpella* spp. survival (classes were grouped by cluster analysis of *Stenocarpella* spp. DNA relative quantifications), soil suppressiveness to *Stenocarpella maydis*, *Fusarium graminearum*, and *Fusarium verticillioides* (% of maize stubble baits colonized), altitude, annual average rainfall, maize yield, and annual average temperature (°C).

The altitude, rainfall, and temperature maps were very similar to each other, i.e., the higher the altitude, the higher the rainfall and the lower the temperature. Such conditions were not exclusive drivers of the importance of *Stenocarpella*-caused diseases since there were regions where the higher the temperature (>20 °C) was, the higher the inoculum level was, as observed for the Varginha sampled sites (Figures 1 and 4). However, under the same temperature condition, such as at the Piumhi location (Figures 1 and 4), a lower inoculum level was found, and this was not related to the suppressiveness of the soil to the disease.

## 4. Discussion

The rate of survival of *Stenocarpella* spp. on maize stubble during the off-season [18] has been the focus of previous studies on grains and stalks [19] or on seeds [5]. Although other maize plant parts such as cobs or grains may serve as the pathogen reservoir of the pathogen within the stubble, they were not always encountered. Indeed, some of the sampled fields had been harvested for forage, and, as such, the cob, along with the grains and most of the shoot, was taken out from the field and only the basal part of the stalks were left. In this way, we set out to study these dynamics in stalks, which were the substrate for the pathogen present in all sampled areas.

Although such estimates of the pathogen inoculum concentrations in maize stubble have been previously carried out, the exclusive observation of the fungal structures of these pathogens, such as their pycnidia and conidia, may underestimate the pathogen inoculum concentration, since inoculum may also be associated with the plant tissue as dormant mycelia. The contribution of crop management systems to *Stenocarpella* spp. survival using DNA-based quantification was addressed in maize stubble in maize fields for the first time, and, as such, not only conidia but also mycelia and any other fungal structures were recovered from the substrate [4].

In addition, the persistence of maize stubble as stalks has a direct impact on the inoculum importance, and the lifespan of the stubble determines the survival of pathogens [20]. In turn, the factors that govern the decomposition of stubble are the C/N ratio, particle size, lignin content, polyphenols, lignin/N ratio, lignin +N/polyphenol ratio, presence of toxic elements, physical and chemical conditions of the soil, and type of microorganisms present in the soil; therefore, these factors govern the survival of *Stenocarpella* spp. [21], which matches our findings. The stalks are representing substrates on which the substantial pathogen concentrations were encountered. These substrates are rich in lignin [22] and therefore last longer than other plant parts [23]. On the other hand, lignin-rich substrates do not offer an environment that is attractive to antagonistic microbial colonization; as a result, *Stenocarpella* spp. should be encountered at higher concentrations in less decomposed stalks, and, in turn, crop management practices would have an impact on such stubble decomposition.

In this regard, the tillage system and crop rotation are major drivers of stubble decomposition. While a no-till system implies a longer lifespan of stubble, crop rotation with legumes (soybean or common bean) results in the amendment of nitrogen to stubble and accelerates the decomposition rate of straw [8]. Furthermore, the disintegration of stubble increases the surface area of its contact with the soil microbiome; whenever the microbiome encompasses antagonistic communities [24], the stubble offers an environment favorable to biological control with a sustained humidity level, stable temperature, and low incidence of ultraviolet radiation [20]. Other approaches may result in the breakdown of straw, favoring a reduction in the survival of pathogens [25].

Cluster analysis for relative quantification by municipalities showed that nearby municipalities do not necessarily group with each other. Although the pathogens are exposed to similar environmental conditions under different crop management practices, different practices lead to different patterns of pathogen occurrence and soil suppressiveness. The development of suppressive soils is a more cost-effective and efficient alternative to microbe application [26]. If soils are evaluated for their suppressiveness prior to planting,

the results will foster decisions regarding the implementation of *Stenocarpella* spp. management practices, such as plowing, fallowing, or not planting maize for the length of time necessary for the maize stubble to decompose.

The observed suppressiveness differed according to the considered pathogen, i.e., the microbiome of the soil acts in a certain way towards a given pathogen and does not necessarily suppress another pathogen, although some of the studied soils could suppress all tested pathogens [24]. The heat maps generated showing the suppressiveness between *Fusarium* species are more similar to each other than to the map generated for the suppressiveness of *Stenocarpella*.

Additionally, the map showing the suppressiveness of *Stenocarpella* spp. and other pathogens does not always match with the concentration of *Stenocarpella* spp. inoculum (Figure 4). The suppressive soils to *Stenocarpella maydis* is an important indication of the lower pathogen inoculum level of *Stenocarpella* spp., but other disease management practices that do not result in suppressive soils are also resulting in the lower pathogen inoculum level (Table 1), and this may be related to the frequency into which the corn is planted within the season and considering maize is the only host of the pathogen not having maize and/or lower maize stubble would result in lower pathogen inoculum. Furthermore, if no pathogen is present, there is a lower chance of selecting for an antagonistic microbial community [27].

The maize yield was also not necessarily explained by the *Stenocarpella* spp. inoculum concentration, suppressiveness of the soil to the different stubble-borne pathogens, or environmental conditions. Although all these factors are reported to be related to maize yield, since each grower undertakes a different combination of the factors, different yields may be obtained. In the maps, it is not possible to clearly determine the relationships between the survival of *Stenocarpella* spp. and maize yield, maize production, or temperature (Figure 4). However, it seems that there are trends among the survival of *Stenocarpella* spp. and the soil suppressiveness, altitude, and precipitation. In areas where more *Stenocarpella* spp. DNA was found, the soil suppressiveness levels were low or absent; this result corroborates the hypothesis that soil suppressiveness can reduce the survival of *Stenocarpella* spp. Areas with high altitudes had higher occurrences of *Stenocarpella* spp. than areas with low altitudes (Figure 4). In [28], when comparing the map classes of *Stenocarpella* spp. survival and precipitation, they concluded that high precipitation favored the survival of *Stenocarpella* spp. [12].

For the integrated management of *Stenocarpella* ssp. in areas where stubble with possibly different inoculum levels are present, farmers use a specific tool to apply fungicide. However, [29] found that the application of fungicide did not consistently reduce white ear rot or improve yield when working with different fungicide spray programs at three different maize growth stages. This failure of fungicides in the management of the disease implies the importance of reducing the inoculum levels of the pathogen in maize stubble [2]. Additionally, glyphosate-resistant, overwintering volunteer maize seedlings [30], which are frequently encountered throughout the year in tropical agricultural fields, represent *Stenocarpella* spp. reservoirs that add to the one already found in the stubble with the advantage of lower competition since many are microorganisms able to colonize the dead plant debris, though fewer have the ability to colonize the live plant.

Therefore, we propose that the distribution of soils that are suppressive to stubble-borne pathogens in tropical soils and *Stenocarpella* inoculum are not necessarily distributed according to the maize growing region or environmental conditions but are more likely to be related to the adopted crop management practices. Although crop rotation and no-till systems were most frequently associated with suppressive soils, these cannot be taken as defaults since there were exceptions and other factors governing both pathogen survival and build-up in the offseason that need to be further dissected.

The pathogen concentration in the stalks was directly associated with the decomposition rate. The crop rotation under no-till systems was associated with soil suppressiveness and reduced pathogen concentrations. Such suppressiveness, when encountered for one

pathogen causing stalk-rot, is not necessarily widespread, but the suppressiveness to *Stenocarpella* spp. is a strong indication of the lower pathogen concentration in the field. Factors other than the antagonistic dominant microbial communities govern the lower concentration of *Stenocarpella* spp.

**Supplementary Materials:** The following supporting information can be downloaded at: https://www.mdpi.com/article/10.3390/app12104974/s1, Table S1: Areas sampled to assess *Stenocarpella* spp. inoculum.

**Author Contributions:** Conceptual Idea: F.H.V.d.M., F.A.M.F.P. and D.D.d.S.; Methodology design: F.H.V.d.M., F.A.M.F.P., J.d.C.M. and D.D.d.S.; Data collection: F.A.M.F.P., V.B.C.P., H.N.M., M.R.d.F. and F.H.V.d.M.; Data analysis and interpretation: F.A.M.F.P., F.H.V.d.M., R.A.G., E.A.P., H.S.N. and C.d.S.S.; Writing and editing: F.A.M.F.P., F.H.V.d.M., R.A.G., V.B.C.P., J.d.C.M. and E.A.P. All authors have read and agreed to the published version of the manuscript.

**Funding:** We thank FAPEMIG (Research Support Foundation of Minas Gerais) grants APQ 02059-13 and APQ01578-15, CNPq (National Council for Research and Development) for the EAP, JCM, and FHVM productivity fellowship (grant 309307/2017-1), CAPES (Coordenação de apoio pessoal de nível superior) and CNPq for the FAMFP, VBCP, RAG, MRF, and HSN scholarships.

**Institutional Review Board Statement:** Not applicable.

**Informed Consent Statement:** Not applicable.

**Data Availability Statement:** Not applicable.

**Acknowledgments:** The authors would like to thank Jürgen Köhl, Pierter Kastelein, and Helen Goossen van de Geijn from Wageningen University and Research Center for technical support on the DNA extraction and qPCR protocol for the *Stenocarpella* quantification, as well as all growers for allowing samples to be collected in their fields.

**Conflicts of Interest:** The authors declare no conflict of interest. The funders had no role in the design of the study; in the collection, analyses, or interpretation of data; in the writing of the manuscript, or in the decision to publish the results.

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
