# Peer review of "Detection and Factors That Induce Stenocarpella spp. Survival in Maize Stubble and Soil Suppressiveness under Tropical Conditions"

_applsci, doi:10.3390/app12104974_

Round 1

Reviewer 1 Report

Feedback

Abstact

  1. Line 2: Replace “inducing” with “induce”.
  2. Line 17: Remove the semi-colon at the end.
  3. Line 19: colonization is crucial strategy
  4. Line 20: Replace “Thus, “ with “In this study”
  5. Line 21: Replace: with the inoculum > to the inoculum.
  6. Line 22-23: Remove: Heat maps were generated based on all data collected.
  7. Line 23: In the 28 fields sampled, different levels of Stenorcapella Stenocarpella spp
  8. Line 28: … tillage systems..
  9. Line 29-30:The sentences are not flowing well, please consider rewriting.

Introduction

  1. Line 37: Replace: after harvest > post-harvest
  2. Line 42-43: Rewrite: In order for an outbreak to occur, the inoculum levels in the maize stubble needs to reach sufficient levels.
  3. Line 45: replace “in”with “using”
  4. Line 45: replace “A molecular biology”
  5. Line 46: What do you mean by ‘more sensibility..’
  6. Line 47-48: Replace with: When more than one pathogen causes infection in a plant and require accurate detection, qPCR is a suitable method for detection and quantification.
  7. Line 51-55: This is all one sentence.Please shorten.
  8. Line 59: Replace ‘maize plant tissues’ with ‘nutrient source’.
  9. Line 59-61:Please clarify the sentence, as it is not clear what the meaning is.
  10. Line 66: Rewrite to clarify: ‘soil biological properties is a direct source of antagonistic microbial communities’. Soil biological properties cannot be a source.  Rather, is can influence the selection of antagonistic microbial communities.
  11. Line 67: replace ‘associated to the pathogen elimination and reduction which results in the reduction of the severity of the diseases they cause’ with ‘associated with the pathogen reduction and elimination, resulting in the reduction of the disease.
  12. Line 68: The rest of the paragraph is out of context and should be given more context.
  13. Line 71: add In Brazil, no-tillage cropping is the default practice
  14. Line 74: Rewrite: ‘that have saprophytic survival abilities and cause disease epidemic outbreaks’ >’ which are saprophytes and can cause disease outbreaks’
  15. Line 76: replace ‘There is considerable knowledge on’ > There is a considerable amount of knowledge’
  16. Line 79: remove ’of pathogen survival’
  17. Line 80: risk of fungal infections
  18. Line 81-84: Rewrite: ‘The objectives of this study were to identify the source of inoculum for Stenocarpella, the contribution of maize growing areas and crop rotation on the Stenocarpella spp., as well as to understand the suppressiveness of soil towards S. maydis, F. verticillioides and F. graminearum’

Materials and Methods

  1. Line 87: Rewrite: Identification of the inoculum source for Stenocarpella spp. within maize stubble
  2. Line 91: Please clarify: Did you make a composite sample of the 5 different potential sources, or to you make a composite of different sources (ie stalks together, grain together etc)?
  3. Line 91: Which species of weed?
  4. Line 92: Rewrite: For each composite sample, the DNA was extracted and relative quantification of Stenocarpella was conducted.
  5. Line 95: Please expand.Did you assume that decomposition was the result of only Stenocarpella?  How did you allow for interactions by other members of the microbiome?
  6. Line 111: Provide a reference.
  7. Line 122: The primers were most likely designed for S. maydis specificity and can be used to calculate DNA copy nr.
  8. Line 143: Was this rhizosphere of bulk soils?
  9. Line 149: Did you use qPCR?How do you correct for the interaction of other microorganisms with the pathogens?
  10. Line 174-181: Remove

Results

  1. Line 92: Stenocarpella should be italics
  2. Line 199-201: Please rewrite the whole paragraph
  3. Line 208-209:I am not sure how you get to this result.  How did you calculate the relation between particle size and pathogen survival?
  4. Line 210-214: How did you calculate and define this?
  5. Line 227: Where did you report the results?
  6. Line 265:From what I see, there is no significant correlation.  I am also confused with what you are trying to measure.
  7. Line 268: The heat maps are only based on a few observations.More samples would give a better idea of the incidence.
  8. Line 288-291: Formatting

Discussion

  1. Line 294-296: remove
  2. Line 297-301: Rewrite
  3. Line 313-318: This was not measured and any correlation with reports in literature should be make with caution.
  4. Line 329-334: Nowhere in the study, the contribution of the larger microbiome has been incorporated.This could potentially have a major influence on pathogen presence and the expression of pathogenicity.
  5. Line 336: No till practices were identified for most of the samples (27/29) with only 2 samples representing tillage (2/29). I would be very cautious about the interpretation of this dataset.

References:

  1. Line 433-436: remove.

Author Response

Dear Reviewer we agreed with all comments made, all modifications in the article are marked with yellow and we provided a answer point-by-point below. 

Line 2: Replace “inducing” with “induce”.

Done.

Line 17: Remove the semi-colon at the end.

Done.

Line 19: colonization is crucial strategy

We replaced this sentence by “colonization is a crucial strategy” as suggested in PDF guides.

Line 20: Replace “Thus, “ with “In this study”

Done.

Line 21: Replace: with the inoculum > to the inoculum.

Done.

Line 22-23: Remove: Heat maps were generated based on all data collected.

Removed.

Line 23: In the 28 fields sampled, different levels of Stenorcapella Stenocarpella spp

Done.

Line 28: … tillage systems.

Done.

Line 29-30:The sentences are not flowing well, please consider rewriting.

The sentence was rewritten.

Introduction

Line 37: Replace: after harvest > post-harvest

Done.

Line 42-43: Rewrite: In order for an outbreak to occur, the inoculum levels in the maize stubble needs to reach sufficient levels.

We reewritted with as suggested.

Line 45: replace “in”with “using”

Done.

Line 45: replace “A molecular biology”

Done

Line 46: What do you mean by ‘more sensibility..’

In this case, the sensibility is in the sense to detection the low DNA concentration in a specific target. The same conditions (low concentrations), the others techniques no has accurated in a detection.

Line 47-48: Replace with: When more than one pathogen causes infection in a plant and require accurate detection, qPCR is a suitable method for detection and quantification.

We rewrite with as suggested.

Line 51-55: This is all one sentence.Please shorten.

We rewrite the sentences and divided for more easy understand.

Line 59: Replace ‘maize plant tissues’ with ‘nutrient source’

Done.

Line 59-61:Please clarify the sentence, as it is not clear what the meaning is.

We clarified the sentence.

Line 66: Rewrite to clarify: ‘soil biological properties is a direct source of antagonistic microbial communities’. Soil biological properties cannot be a source.  Rather, is can influence the selection of antagonistic microbial communities.

We rewrite with as suggested.

Line 67: replace ‘associated to the pathogen elimination and reduction which results in the reduction of the severity of the diseases they cause’ with ‘associated with the pathogen reduction and elimination, resulting in the reduction of the disease.

Done.

Line 68: The rest of the paragraph is out of context and should be given more context.

We rewrite the all sentences and linked with the last paragraph.

Line 71: add In Brazil, no-tillage cropping is the default practice

Done.

Line 74: Rewrite: ‘that have saprophytic survival abilities and cause disease epidemic outbreaks’ >’ which are saprophytes and can cause disease outbreaks’

We rewrite with as suggested.

Line 76: replace ‘There is considerable knowledge on’ > There is a considerable amount of knowledge’

In this sentence, we replaced “knowledge” by “references”.

Line 79: remove ’of pathogen survival’

Done.

Line 80: risk of fungal infections

Done.

Line 81-84: Rewrite: ‘The objectives of this study were to identify the source of inoculum for Stenocarpella, the contribution of maize growing areas and crop rotation on the Stenocarpella spp., as well as to understand the suppressiveness of soil towards S. maydis, F. verticillioides and F. graminearum’

We rewrite with as suggested.

Materials and Methods

Line 87: Rewrite: Identification of the inoculum source for Stenocarpella spp. within maize stubble.

We rewrite with as suggested.

Line 91: Please clarify: Did you make a composite sample of the 5 different potential sources, or to you make a composite of different sources (ie stalks together, grain together etc)?

We clarified the sentence. In this case, we used of different sources (stalks, grains, cobs, decaying maize leaves and dead weeds )in different fields (composite sample is derivated of fields) for checking if have the specific source of Stenocarpella spp. inoculum survive.

Line 91: Which species of weed?

The species of weed were not classified.

Line 92: Rewrite: For each composite sample, the DNA was extracted and relative quantification of Stenocarpella was conducted.

Done

Line 95: Please expand.Did you assume that decomposition was the result of only Stenocarpella?  How did you allow for interactions by other members of the microbiome?

The decomposition was the result of integrated de cropping systems and the microbiome that are involved in this conditions, in this case cropping rotation and not manocropping is the best way for reduction the Stenocarpella spp inoculum level.

Line 111: Provide a reference.

Done

Line 122: The primers were most likely designed for S. maydis specificity and can be used to calculate DNA copy nr.

Agreed.

Line 143: Was this rhizosphere of bulk soils?

Yes, we insert this information!

Line 149: Did you use qPCR?How do you correct for the interaction of other microorganisms with the pathogens?

No, by the level of colonization of the baits with the evaluated pathogens.

Line 174-181: Remove

Done.

Results

Line 92: Stenocarpella should be italics

Done.

Line 199-201: Please rewrite the whole paragraph

Done

Line 208-209:I am not sure how you get to this result.  How did you calculate the relation between particle size and pathogen survival?

Checking the size and comparing with the pathogen survival.

Line 210-214: How did you calculate and define this?

According to one-way ANOVA and Tukey’s multiple comparisons of means, stubble particle size contributed to pathogen survival, as measured by the DNA quantification of Stenocarpella spp. The relative quantification of stalks classified as entire was different from those classified as disintegrated and was equal to those classified as partially disintegrated. Areas with disintegrated stalks were different from areas with partially disintegrated stalks.

Line 227: Where did you report the results?

In the Table 1, we putted a call for this information.

Line 265: From what I see, there is no significant correlation.  I am also confused with what you are trying to measure.

In this case, we liked checked if in a supressiviness for a pathogens can extended for other, thus checked all sampled and the fields for this conditions.

Line 268: The heat maps are only based on a few observations. More samples would give a better idea of the incidence.

Sure, always that can be more sampling points is more specific and precision. However, in this work, we suggested the potential of Stenocarpella spp. survive in stubble and this relationship with crop systems in tropical conditions.

Line 288-291: Formatting

Done.

Discussion

Line 294-296: remove

Done.

Line 297-301: Rewrite

Done

Line 313-318: This was not measured and any correlation with reports in literature should be make with caution.

We reorganized the order of the paragraphs in order to facilitate the discussion.

Line 329-334: Nowhere in the study, the contribution of the larger microbiome has been incorporated.This could potentially have a major influence on pathogen presence and the expression of pathogenicity.

It is true, but is important discuss about the influence of larger microbiome on pathogen presence

Line 336: No till practices were identified for most of the samples (27/29) with only 2 samples representing tillage (2/29). I would be very cautious about the interpretation of this dataset.

Agreed, however it is not easy find samples for tillage, since almost all farmers are using only no till pratices.

References

Line 433-436: remove.

Done.

Reviewer 2 Report

In this article, abiotic factors and crop management practices are analyzed in relation to the inoculum of Stenocarpella spp. in stubble by qPCR........
The work from my point of view is correct, but some clarifications or improvements must be reviewed or clarified prior to the acceptance of the article.

1. In the summary there are 28 fields and in line 102 it is said of 29 samples collected from 15 different populations (Figure 1). However, in Table 1 and Table S1, 29 fields are shown and not 28. We believe that it is necessary to clarify this mismatch and specify whether there are 28 or 29 fields under study.

2.- When P1 and P2 are introduced in the qPCR analysis for the first time in lines 120 and 121, they must be explained in a little more detail as described in references (3 and 5) and the measurements of the generated fluorescence and how the threshold Ct and RQ curves are determined and their meaning, according to reference (14). We estimate that Figure 2 needs to be explained in more detail as it has been calculated and make the three sections A), B) and C) the same size of figure and letter and explain what CT and P1/P2 mean without having to go to the text.

3.- We believe that lines 278 to 285 should be rewritten in more detail, so that the reader can better understand the color maps and the dependence of the soil and temperature factor on the degree of disease. We recommend that the authors mention and explain again the color scale for the two examples from Varginha and Piumhi (Figures 1 and 4).

4.- It is not understood at the beginning of the discussion that "the authors" are the ones who should discuss.... and I should say the scholars or readers of the subject. Please review lines 294 to 300 again for a better understanding by readers.

Author Response

Dear Reviewer we agreed with all comments made, all modifications in the article are marked with yellow and we provided a answer point-by-point below.

In this article, abiotic factors and crop management practices are analyzed in relation to the inoculum of Stenocarpella spp. in stubble by qPCR........
The work from my point of view is correct, but some clarifications or improvements must be reviewed or clarified prior to the acceptance of the article.

  1. In the summary there are 28 fields and in line 102 it is said of 29 samples collected from 15 different populations (Figure 1). However, in Table 1 and Table S1, 29 fields are shown and not 28. We believe that it is necessary to clarify this mismatch and specify whether there are 28 or 29 fields under study.

Agreed. The correct number of fields is 29, we modified in summary.

2.- When P1 and P2 are introduced in the qPCR analysis for the first time in lines 120 and 121, they must be explained in a little more detail as described in references (3 and 5) and the measurements of the generated fluorescence and how the threshold Ct and RQ curves are determined and their meaning, according to reference (14). We estimate that Figure 2 needs to be explained in more detail as it has been calculated and make the three sections A), B) and C) the same size of figure and letter and explain what CT and P1/P2 mean without having to go to the text.

Agreed.

3.- We believe that lines 278 to 285 should be rewritten in more detail, so that the reader can better understand the color maps and the dependence of the soil and temperature factor on the degree of disease. We recommend that the authors mention and explain again the color scale for the two examples from Varginha and Piumhi (Figures 1 and 4).

Agreed.

4.- It is not understood at the beginning of the discussion that "the authors" are the ones who should discuss.... and I should say the scholars or readers of the subject. Please review lines 294 to 300 again for a better understanding by readers.

Agreed. We modified the text in order to improve the quality.

Reviewer 3 Report

Dear Authors,

I think that your manuscript is adequately conceived and organized in each section opening to a clear and actual insight of the topics. The quality of the paper is very high and my impression is positive and, in my opinion paper fully accomplished the scope of Applied Sciences journal. Comprehensively, the paper include new and very interesting data and it should be published.

For other minor comments/suggestions, I provide in attachment an annotated PDF.

Based on above reported, I believe manuscript should be accepted with a few revisions.

Author Response

Dear reviewer, we thank for the comments. We agree with all comments. All the changes in the text were marked with yellow. We did not use the arcsine square transformation because was not necessary in that case.

Round 2

Reviewer 1 Report

Feedback (review2)

  1. Line 67-71: Rewrite to clarify: ‘soil biological properties is a direct source of antagonistic microbial communities’. Soil biological properties cannot be a source.  Rather, it can influence the selection of antagonistic microbial communities. ( I am not sure what you mean, I can see the rewrite, but I do not understand the meaning.)
  2. Line 77: replace ‘There is considerable knowledge on’ > There is a considerable amount of knowledge’(Amount: if you cannot count vs number: where you can count it)

Materials and Methods

  1. Line 93: Please clarify: Did you make a composite sample of the 5 different potential sources, or to you make a composite of different sources (ie stalks together, grain together etc)? Please work your explanation into the text, it is not clear from reading it.
  2. Line 93: Which species of weed? (work into the text)
  3. Line 97: Please expand.Did you assume that decomposition was the result of only Stenocarpella?  How did you allow for interactions by other members of the microbiome?  (You did not answer this question.)
  4. Line 125: The primers were most likely designed for S. maydis specificity and can be used to calculate DNA copy nr. (still require clarification.)
  5. Line 149: Did you use qPCR? How do you correct for the interaction of other microorganisms with the pathogens? (still require clarification.)

Results

  1. Line 196-205: Please rewrite the whole paragraph (not done)
  2. Line 208-209:I am not sure how you get to this result.  How did you calculate the relation between particle size and pathogen survival? (Please work this into the text)
  3. Line 210-214: How did you calculate and define this? (You have not answered this, copying the text into the reply, does not constitute an explanation)

Author Response

Dear Reviewer, all the comments were accepted, and the manuscript was revised. The changes made in the document were performed using marked text (blue). All the comments are answered point-by-point below.

  1. Line 67-71: Rewrite to clarify: ‘soil biological properties is a direct source of antagonistic microbial communities’. Soil biological properties cannot be a source.  Rather, it can influence the selection of antagonistic microbial communities. ( I am not sure what you mean, I can see the rewrite, but I do not understand the meaning.)

Agreed. We modified the text in order to clarify.

2. Line 77: replace ‘There is considerable knowledge on’ > There is a considerable amount of knowledge’(Amount: if you cannot count vs number: where you can count it)

Agreed.

Materials and Methods

3. Line 93: Please clarify: Did you make a composite sample of the 5 different potential sources, or to you make a composite of different sources (ie stalks together, grain together etc)? Please work your explanation into the text, it is not clear from reading it.

Agreed. We collected each type of sample (e.g. stalks together) and tested each type in order to verify major sources of inoculum.

4. Line 93: Which species of weed? (work into the text)

Agreed. The species of weed were not identified.

5. Line 97: Please expand.Did you assume that decomposition was the result of only Stenocarpella?  How did you allow for interactions by other members of the microbiome?  (You did not answer this question.)

We creating a subsection in materials and methods to better explain this part of the work. The decomposition is resulted by microbiome and environment. Stenocarpella uses the stubble to survive. We collected the soil samples and inoculated them with the pathogens to see the supressiveness.

6. Line 125: The primers were most likely designed for S. maydis specificity and can be used to calculate DNA copy nr. (still require clarification.)

The quantification was actually for Stenocarpella sp. The used primer set does not make a distinction between Stenocarpella maydis and Stenocarpella macrospora and the reason why we used this primer set is because we have actually both species reported in our region as associated to maize stubble and causing damage to the crop. Therefore, from the performed qPCR we can compare samples for the cycle threshold but we cannot infer the amount of DNA we in each sample since we could not distinguish between species. The determination of the population of each species and the damage each one is causing is object of future research our group envisage but by itself would be another PhD thesis.

7. Line 149: Did you use qPCR? How do you correct for the interaction of other microorganisms with the pathogens? (still require clarification.)

The Suppressiveness was tested using bait colonization method, not qPCR. We collected soil from each site (29 fields), inoculated them with 3 maize pathogens separated and evaluated the number of baits colonized.

Results

8. Line 196-205: Please rewrite the whole paragraph (not done)

Agreed. We modified the text in order to clarify the results.

9. Line 208-209:I am not sure how you get to this result.  How did you calculate the relation between particle size and pathogen survival? (Please work this into the text)

We worked in the text, in materials and methods, creating a subsection to better explain this part of the work. We compared the pathogen survival from each site (by qPCR) with the classification of stubble decomposition from each site (high, medium or low level). (line 99-106)

10. Line 210-214: How did you calculate and define this? (You have not answered this, copying the text into the reply, does not constitute an explanation)

We added one sentence in order to clarify in the text. (line 147). We define this and calculated using the formula as described in the materials and methods: the Ct values (exponential increase in PCR product) of the target gene and endogenous control, ∆Ct = Ct (sample) - Ct (endogenous control) and ∆∆Ct = ∆Ct (sample) - ∆Ct (calibrator). The gene expression levels were then calculated using the formula RQ = 2-∆∆Ct, where RQ means relative quantifications.

Round 3

Reviewer 1 Report

No comments as the issues have not been addressed to the satisfaction of the reviewer. See previous review